# Magnetic Resonance Imaging, Computed Tomographic and Radiographic Findings in the Metacarpophalangeal Joints of 40 Non-Lame Thoroughbred Yearlings

**DOI:** 10.3390/ani13223466

**Published:** 2023-11-09

**Authors:** Annamaria Nagy, Koppány Boros, Sue Dyson

**Affiliations:** 1Equine Department and Clinic, University of Veterinary Medicine Budapest, Doramajor, 2225 Ullo, Hungary; boros.koppany@univet.hu; 2The Cottage, Church Road, Market Weston, Diss IP22 2NX, UK; sue.dyson@aol.com

**Keywords:** fetlock, racehorse, imaging, stress-related injury, osteochondrosis, developmental abnormalities, sclerosis, adaptive modelling

## Abstract

**Simple Summary:**

Most catastrophic musculoskeletal injuries in Thoroughbred racehorses involve the fetlock region. Magnetic resonance imaging (MRI) and computed tomography (CT) aid detection of potentially serious abnormalities, before they become identifiable on radiographs. However, no data exist in live horses regarding how these injuries develop and progress. The aim of this study was to describe MRI, CT and radiographic findings in the front fetlock of Thoroughbreds entering racehorse training. Forty non-lame yearlings underwent low-field MRI, CT and radiographic examinations of both front fetlocks. The most common finding was a lesion consistent with osteochondrosis (developmental abnormality) in the cannon bone. Of the 33 lesions seen in CT images, about two-thirds were detected on radiographs. In the majority of the cannon bones, there was some increase in bone density and enlargement of vascular channels, consistent with the bone’s response to exercise. Small defects in the joint surface were seen in both the cannon bone (11 limbs) and the long pastern bone (19 limbs) which may indicate developmental abnormalities or subtle injuries. In conclusion, variants or abnormalities occur in the cannon bone in the fetlock region of yearling Thoroughbreds and may reflect early changes as the bone adapts to exercise or changes secondary to osteochondrosis.

**Abstract:**

Most catastrophic injuries in Thoroughbred racehorses involve the fetlock. There is no description of comparative imaging in Thoroughbreds entering racehorse training. The aim was to describe MRI, CT and radiographic findings in the metacarpophalangeal joint of non-lame Thoroughbred yearlings. Forty Thoroughbreds underwent low-field MRI, fan-beam CT and radiographic examinations of both metacarpophalangeal joints. Images were assessed subjectively. A hypoattenuating lesion of the sagittal ridge of the third metacarpal bone (McIII) was identified in 33/80 limbs in CT reconstructions. Cone-shaped mineralisation in the sagittal ridge was detected in MR images (*n* = 17) and in CT images (*n* = 5). Mild hyperattenuation was common in trabecular bone in the dorsomedial (36/80) and palmarolateral (25/80) metacarpal condyles in CT reconstructions. A focal lesion in the subchondral bone was seen in the proximal phalanx (*n* = 19) and in McIII (*n* = 11). Enlarged vascular channels were detected in the metacarpal condyles in 57/80 limbs and in the proximal sesamoid bones in all limbs. Signs of bone modelling are seen in yearling Thoroughbred fetlocks. Sagittal ridge lesions were common and are likely associated with osteochondrosis or other developmental osteochondral defects. Focal lesions in the subchondral bone of McIII and proximal phalanx can indicate developmental abnormalities or subtle subchondral bone injuries.

## 1. Introduction

Metacarpophalangeal joint region pain is one of the most common causes of lameness in Thoroughbred racehorses, and this region is the predilection site of catastrophic racehorse injuries worldwide [1,2,3,4,5]. Non-adaptive stress-related injuries and prodromal lesions of potentially catastrophic fractures in the metacarpophalangeal joint have been documented, most commonly in the metacarpal condyles, in the proximal sesamoid bones and in the dorsoproximal aspect of the proximal phalanx [1,2,6,7,8,9,10,11]. There is no in vivo evidence-based information on the progression of adaptive osseous changes into non-adaptive pathological lesions. Diagnostic imaging studies of racehorses prior to entering training and racing are limited to presale radiographs [12,13,14]. To our knowledge, there are no published data on lesions detectable using magnetic resonance imaging (MRI) and/or computed tomography (CT) in yearling Thoroughbreds entering racehorse training. These advanced diagnostic imaging modalities are increasingly used in the investigation of metacarpophalangeal joint pain and have been useful for detecting stress-related pathology and prodromal lesions of potentially catastrophic injuries [2,15,16,17,18]. It is therefore imperative that adaptive changes and other abnormalities already present in the early stages of racehorse training are described. The overall aim of this study was to document low-field MRI, CT and radiographic findings in the metacarpophalangeal joints of non-lame Thoroughbred yearlings. We aimed to provide reference data on abnormalities that are already present in young Thoroughbreds with no prior intensive training.

## 2. Materials and Methods

The current study is the first part of a longitudinal project in which sequential comparative imaging of the metacarpophalangeal joints is being performed in Thoroughbred racehorses from the time of entering racehorse training as a yearling to the end of the horses’ third year of age. 

Thoroughbred racehorse trainers were invited to provide horses for the study; participation was voluntary. Horses born in the previous year, free from lameness and with no history of metacarpophalangeal joint disease were eligible. A clinical history was obtained, including the start of training, exercise routine (if any) and any previous lameness. Clinical and lameness examinations (subjective and objective (Equinosis Q Lameness Locator, SC, USA) lameness assessment in a straight line) were performed. The first 40 horses that did not show overt lameness in walk or trot were selected. All diagnostic imaging of both metacarpophalangeal joints was performed in a standing position with the horses sedated using a combination of acepromazine (Vetoquinol, Lure, France) (0.04 mg/kg IM), romifidine (Boehringer Ingelheim, Ingelheim, Germany) (0.01 mg/kg IV), detomidine (Orion Pharma, Budapest, Hungary) (0.01 mg/kg IV) and butorphanol (Bioveta, Ivanovice na Hané, Czech Republic) (0.02 mg/kg IV). Low-field MRI was performed in a 0.27 T open magnet (Hallmarq Veterinary Imaging Ltd., Guildford, UK); T1-weighted and T2*-weighted gradient echo and short-tau inversion recovery sequences in sagittal, frontal and transverse planes, and T2-weighted fast spin echo sequences in transverse planes, were acquired (Table 1). 

Fan-beam computed tomography was performed using a 16-detector multislice helical scanner with a 90 cm gantry and acquisition values of 135 kV and 350 ms, 300 mm range, 300 mm field of view, 0.5 mm slice thickness and 0.5 sec rotation time (Canon Aquilion LB, Qalibra, CT, USA). Each limb was scanned from the distal third of the metacarpal region to the level of the proximal interphalangeal joint. Radiographic examination included standard (lateromedial, dorsal 10° proximal–palmarodistal oblique, dorsal 45° lateral–palmaromedial and palmar 45° lateral–dorsomedial oblique) and additional (lateromedial (flexed), dorsopalmar (flexed), dorsal 70° proximal 45° lateral–palmarodistomedial oblique and dorsal 70° proximal 45° medial–palmarodistolateral oblique and dorsoproximal–dorsodistal (flexed) (skyline)) views (Fuji DR-ID1270, Fujifilm Corporation, Tokyo, Japan). All diagnostic imaging acquisitions were performed within 30 h.

Images were assessed subjectively using medical viewing software (JiveX DICOM Viewer^®^ Version 5.2, Visus Health IT GmbH, Bochum, Germany) following a protocol adapted to each imaging modality (Table 2). Presence and location of abnormalities in the cortical, subchondral, trabecular and compact bone and in soft tissues were recorded. Computed tomographic images were reviewed using multiplanar reconstructions with bone and soft tissue algorithms [17]. Hyperattenuation in the trabecular bone was defined as an increase in attenuation and loss of architecture. The pattern of subchondral bone thickness was described in sagittal CT reconstructions in the middle of the medial glenoid of the proximal phalanx, at the deepest point of the sagittal groove and in the middle of the lateral glenoid, as uniform, thicker dorsally or thicker in the palmar aspect. Thickening and symmetry of the subchondral bone were assessed in frontal reconstructions in the dorsal, middle and palmar thirds of the proximal phalanx. In order to determine whether the subchondral bone was thickened, an imaginary line was drawn tangential to the articular surface of the medial and lateral glenoids; a second line, parallel to the first line, was drawn at the distal extent of the articular surface of the sagittal groove of the proximal phalanx. The subchondral bone was considered thickened if it extended to the level of or beyond this line (Figure 1). Conclusions were reached by consensus of two (MRI) (an Associate of the European College of Veterinary Diagnostic Imaging (S.D.) and a Diplomate of the European College of Veterinary Sports Medicine and Rehabilitation (A.N.)) or all three authors (CT and radiography). The junior author (K.B.) is a PhD student receiving special training in cross-sectional imaging. 

## 3. Results

All examinations took place within a seven-week period in November–December 2021. Horses’ ages ranged from 536 to 680 days (mean 612.3, median 612). There were 26 colts and 14 fillies. Twenty-seven horses had been engaged in some form of training (for 3–16 weeks, mean 7.4, median 4); 26 were exercised in trot and one also in canter (no fast work). Minimal distension of the metacarpophalangeal joint was palpated in four horses, in two horses in one limb and in the other two bilaterally. No other significant abnormalities related to the metacarpophalangeal joint were detected in any horse.

No horses showed overt forelimb lameness in walk or trot in a straight line repeatedly. The results of objective lameness assessment were deemed unreliable. Horses were unable to trot in a straight line maintaining a consistent straight direction, with the head in alignment with the trunk, and for a sufficient number of consecutive steps that would be adequate for analysis. Distal limb flexion tests could not be performed in the majority of horses. Radiographic and CT studies were available in full for all horses. In one horse, no diagnostic-quality MR images could be acquired due to excessive movement. 

### 3.1. The Third Metacarpal Bone (McIII)

A hypoattenuating lesion in the dorsal subchondral bone of the sagittal ridge of McIII was recorded in 33/80 (41.3%) limbs in CT reconstructions (Figure 2). In 22/32 (68.8%) limbs (for one limb with CT abnormality, no MR images were available), there was associated focal hyperintense signal in all MR sequences. Of the 33 hypoattenuating lesions, 25 (75.6%) were detected on radiographs, 13 in standard views and 12 in additional views only (LM (flexed), *n* = 12, DPa (flexed), *n* = 5, skyline, *n* = 5).

There was a cone-shaped pattern of low signal intensity widening towards the dorsodistal aspect of the sagittal ridge in sagittal T1- and T2*-weighted GRE sequences of 19/40 (47.5%) limbs (Figure 2). In 12 of these limbs (62.3%), there was a hypoattenuating lesion in the dorsal subchondral bone of the sagittal ridge in CT images. A cone-shaped pattern of hyperattenuation was also recognised in 5/19 (26.3%) limbs in CT reconstructions; in the remaining limbs, patchy hyperattenuation was seen in the trabecular bone. 

In CT images, mild dorsal subchondral bone thickening in the sagittal ridge was detected in 74/80 (92.5%) limbs (Figure 3). Mild subchondral bone thickening in the palmar aspect of the sagittal ridge was only seen in one horse (both limbs). A small, mineralised opacity/hyperattenuation was identified on radiographs and CT reconstructions just dorsal to the proximal aspect of the sagittal ridge in two limbs; neither was visible on MR images. 

In CT reconstructions, mild hyperattenuation suggesting increased bone density was most common in the dorsal aspect of the medial condyle (36/80 limbs, 45.0%) and in the palmar aspect of the lateral condyle (25/80 limbs, 31.3%) (Figure 4). In MR images, the signal intensity changes were not considered marked enough to allow consistent and reliable subjective judgement on the decrease in signal intensity.

A small hypoattenuating region was identified in the subchondral bone of the medial condyle (in or just medial to the parasagittal groove) in five limbs (Figure 5) and in the lateral condyle in six limbs (five in the parasagittal groove and one halfway between the axial and abaxial aspects of the condyle). In only one limb was the hypoattenuating lesion surrounded by increased attenuation. There was also focally increased attenuation in the articular cartilage and correspondingly altered MR signal intensity (focal increase in the subchondral bone, surrounded by low signal intensity) (Figure 6). In no other limbs were the hypoattenuating lesions identified in MR images or radiographs. Enlarged vascular channels originating from the palmar aspect of the parasagittal groove were detected in the medial condyle in 57/80 limbs (71.3%) and in the lateral condyle in 49/80 limbs (61.3%) (Figure 7). 

### 3.2. The Proximal Phalanx

In the majority of limbs (52/80, 65.0%), the subchondral bone plate of the sagittal groove was thickest in the middle of the weight-bearing surface. In 21/80 (26.3%) limbs, the thickness was uniform, in five limbs it was thicker dorsally and in four in the palmar aspect. In both glenoids of the proximal phalanx, the subchondral bone plate was thicker in the palmar aspect than dorsally (medial: 50/80, 62.5%; lateral: 55/80, 68.8%). In the remaining limbs, the subchondral bone plate had uniform thickness from dorsal to palmar in both glenoids. Marked thickening of the subchondral bone or increased density in the trabecular bone was not seen in any limb. In the majority of the limbs, the subchondral bone thickness was symmetrical between the medial and the lateral glenoid in the dorsal and mid third of the bone. Dorsally, the subchondral bone was mildly thicker medially compared with laterally in one limb and laterally compared with medially in 11 limbs. In the middle of the weight-bearing surface, the subchondral bone was mildly thicker medially compared with laterally in two limbs and laterally compared with medially in three limbs. In the palmar third of the proximal phalanx, the subchondral bone of the medial glenoid was thicker than the lateral glenoid in 12 limbs and thicker in the lateral glenoid compared with medially in 33 limbs.

A smooth indentation in the articular surface was identified in the sagittal groove of the proximal phalanx in eight limbs and in the medial (*n* = 2) or lateral (*n* = 1) parasagittal groove in three limbs in CT reconstructions (Figure 8); none was detected in MR images or radiographs. There was an irregular hypoattenuating lesion surrounded by focal subchondral bone thickening in the sagittal groove in four limbs (Figure 9) and just medial (*n* = 2) or lateral (*n* = 2) to the sagittal groove in four limbs in CT reconstructions. One was identified on radiographs and none in MR images. A small, mineralised opacity/focal hyperattenuation was identified on radiographs and CT reconstructions just proximal to the dorsoproximal aspect of the proximal phalanx in two limbs. 

### 3.3. The Proximal Sesamoid Bones (PSBs)

In CT reconstructions, there was thickening of the abaxial compact bone of the medial PSB in 37/80 (46.3%) limbs (Figure 10) and of the lateral PSB in 17/80 (21.3%) limbs. Prominent vascular channels were identified in most medial PSBs (in all CT images and in 74/80 (92.5%) radiographs) and in the majority of lateral PSBs (74/80 (92.5%) in CT images and 62/80 (77.5%) in radiographs). On radiographs, an irregular palmar surface of the medial PSB was recorded in 13 limbs, and in the lateral PSB in six limbs, which was not recognised in CT or MR images.

### 3.4. Soft Tissues

In CT reconstructions, there was asymmetry in the cross-sectional area of the suspensory ligament branches in 16/80 (20.0%) limbs. The medial branch was larger than the lateral branch in 11 limbs; the lateral branch was larger than the medial branch in six limbs. A hypoattenuating lesion was seen in one lateral suspensory ligament branch. The MRI protocol did not include sequences to evaluate the suspensory ligament branches. In CT images, the oblique sesamoidean ligaments were asymmetrical in size in 26/80 (32.5%) limbs. The medial oblique sesamoidean ligament was larger than the lateral in seven limbs, and the lateral oblique sesamoidean ligament was larger than the medial in 19 limbs. In MR images, the lateral oblique sesamoidean ligament appeared larger than the medial in 10 limbs. No other soft tissue abnormalities were detected.

## 4. Discussion

This is the first study to describe MRI and CT findings in yearling Thoroughbreds entering racehorse training. In agreement with previously published literature [12,13,14], imaging findings suggestive of osteochondrosis were common in the sagittal ridge of McIII. In presale radiographs of 1127 Thoroughbred yearlings, a radiolucent lesion in the distal aspect of the sagittal ridge was reported in 17.4% of horses [12], which is a lower prevalence than presented in the current study. Computed tomographic reconstructions avoid superimposition and thus facilitate lesion detection [19,20]. However, the prevalence of radiographically identified lesions was also higher than previously reported, which may be explained by a population from a geographical region different to that of the current study. Conventional radiography was used by Kane et al. [12], whereas digital radiography was used in the current study, and this may also have contributed to increased detection of lucent lesions [21]. Alternatively, the smaller sample size of the current study may not be representative of a larger population. 

To our knowledge, a cone-shaped pattern of increased density in the trabecular bone of the sagittal ridge of McIII has not been described. In the current study, in the majority of limbs, this was associated with a hypoattenuating lesion in the dorsal subchondral bone of the sagittal ridge in CT images. However, this cone-shaped increase in bone density was observed in horses with and without dorsal sagittal ridge lesions and may reflect modelling either because of dorsal sagittal ridge lesions or as an adaptation to exercise. This pattern of increased bone density was most obvious on MR images, probably because slice thickness was greater for MR images compared with CT images, with signal intensity being averaged across 5 mm, resulting in more obvious changes. The correlation between CT and MR appearance was beyond the scope of this study and requires further investigation. 

Previous studies have established an association between increased bone density and/or histologically confirmed sclerosis and intense exercise [22,23,24,25]. In younger horses, before regular racehorse training, different results have been presented. A longitudinal study used peripheral quantitative CT to describe changes in the third carpal bone, distal aspect of McIII and in the proximal phalanx of Thoroughbreds from 3 weeks to 17 months of age subjected to free pasture exercise or pasture exercise and additional conditioning [26]. Mild exercise early in life resulted in an increase in bone size and strength but not in bone density. This contradicts the results of the current study, where increased bone density was seen in both the sagittal ridge and the condyles of McIII. The mean and median age of the population of the current study was 20 months, and over half of the horses had been engaged in trotting exercise. The cumulative effects of additional age and trotting exercise may explain the difference in results. Moreover, we do not have information on bone density in the study population at an earlier age; therefore, results of the two studies are not directly comparable. In a recent cadaver study, micro-CT was performed on the palmar aspect of the metacarpal condyles of four yearlings and five untrained 2-year-olds [27]. The largest differences in micromorphology of the subchondral bone (bone volume fraction, size of bone pores and tissue mineral density) were observed in untrained horses between the age of 16 and 20 months, which is in accordance with the imaging results of the current study, indicating an increase in bone density. 

An association between increased bone density and the risk of condylar fractures has been described in post-mortem studies [3,4,5,6]. In the current study, diagnostic imaging changes consistent with mild trabecular bone mineralisation (reflecting increased bone density) were mostly seen in the dorsal aspect of the medial and in the palmar aspect of the lateral metacarpal condyles. Enlargement of the vascular channels in the metacarpal condyles was also observed and, together with increased bone density, may likely reflect adaptive changes and bone modelling already occurring in the early stages of training or be a result of free exercise as a foal. While the effect of early exercise on the bone has been described [26], to our knowledge, enlarged vascular channels have not been previously documented.

The smooth surface of indentations in the articular surface of the subchondral bone of the proximal phalanx and McIII and the lack of trabecular bone changes around them suggest that these might be congenital defects. There were also irregularly margined hypoattenuating lesions in the subchondral bone of the proximal phalanx, the significance and aetiology of which are unclear. Follow-up imaging studies are required to establish progression of these abnormalities and whether they present a risk for developing clinically significant lesions (e.g., proximal phalangeal or condylar fractures or palmar osteochondral disease). In live horses, resorptive lesions in the sagittal groove of the proximal phalanx have been associated with subchondral bone trauma and short incomplete fractures in Warmbloods [28], and similar MRI findings have been documented in Thoroughbreds [15]. Recently, histopathological features and comparative low-field MRI, fan- and cone-beam CT findings of fissures in the sagittal groove of the proximal phalanx and in the parasagittal groove of McIII and the third metatarsal bone have been described in 31 limbs of 10 Thoroughbred racehorses aged 1–11 years [29]. The smoothly marginated indentation seen in the current study both in the proximal phalanx and in the McIII do not resemble the lesions described as fissures in this cadaver study. The hypoattenuating subchondral lesions with irregular margins observed in the current study may reflect fissures. In the cadaver study, any association between clinical findings (including age), lameness and the imaging findings was not assessed [29]. The potential presence of subchondral bone fissures in non-lame yearlings needs further investigation, including evaluation of their progression with increasing workload and histological characterisation of the lesions. 

The focal hyperattenuation/hypointense signal in the region of the articular cartilage detected in one limb resembled, but was not identical to, central subchondral osteophytes that have been described on MR and CT images in association with osteoarthritis in racehorses [30,31]. In the horse described in the current study, there was associated subchondral hypoattenuation/hyperintense signal, which was not documented in the previous studies.

Proximal sesamoid bone abnormalities detected radiographically in Thoroughbred yearlings have been associated with decreased racing performance as 2- and 3-year-olds [14,30]. Changes in bone morphology, including increased bone volume fraction, bone width and trabecular thickness, and presence of focal subchondral lesions have been associated with proximal sesamoid bone fractures in cadaver studies using histology and quantitative CT [10,11]. Detailed description of diagnostic imaging findings in the proximal sesamoid bone and their association with other lesions in the metacarpophalangeal joint and with performance will be published elsewhere. 

No overt soft tissue abnormalities were detected in the current study. Mild asymmetry in size was seen between the medial and lateral suspensory ligament branches and oblique sesamoidean ligaments in a small proportion of horses. Asymmetry in size and infrastructure of the suspensory ligament branches has been reported in non-lame older Thoroughbreds in training [32,33]. Asymmetry in the size of the medial and lateral oblique sesamoidean ligaments has been reported without clinical significance [34]. To our knowledge, MRI or CT findings in the suspensory ligaments of non-lame horses have not been documented. 

The main limitations of the study include a relatively small sample size and the inherent limitations of low-field MRI (low resolution and movement artefacts). Although some horses had commenced light training, differentiation between those which had begun ridden exercise and those which had not would have a resulted in a study with low statistical power with potentially misleading results. Moreover, variability in exercise patterns during paddock turn-out could not have been accounted for. Pre-training exercise may contribute to adaptive osseous changes [27].

Studying live horses prohibited histological evaluation of the described lesions. The association between detected abnormalities and cumulative exercise history and the assessment for sensitivity and specificity of each imaging modality for lesion detection are beyond the scope of this descriptive study and will be investigated separately for each type of lesion, including objective assessment. Conformation may have had an influence on our findings. No horses had any major conformation abnormality, but association between low-grade conformational abnormalities and imaging findings was not assessed. 

## 5. Conclusions

In conclusion, early adaptive trabecular bone changes in the metacarpal condyles and more marked increased bone density in the sagittal ridge of McIII, with and without lesions consistent with osteochondrosis, can be seen in non-lame yearling Thoroughbreds entering racehorse training. Hypoattenuating lesions in the subchondral bone of McIII and proximal phalanx may indicate developmental abnormalities or subtle subchondral bone injuries.

## Figures and Tables

**Figure 1 animals-13-03466-f001:**
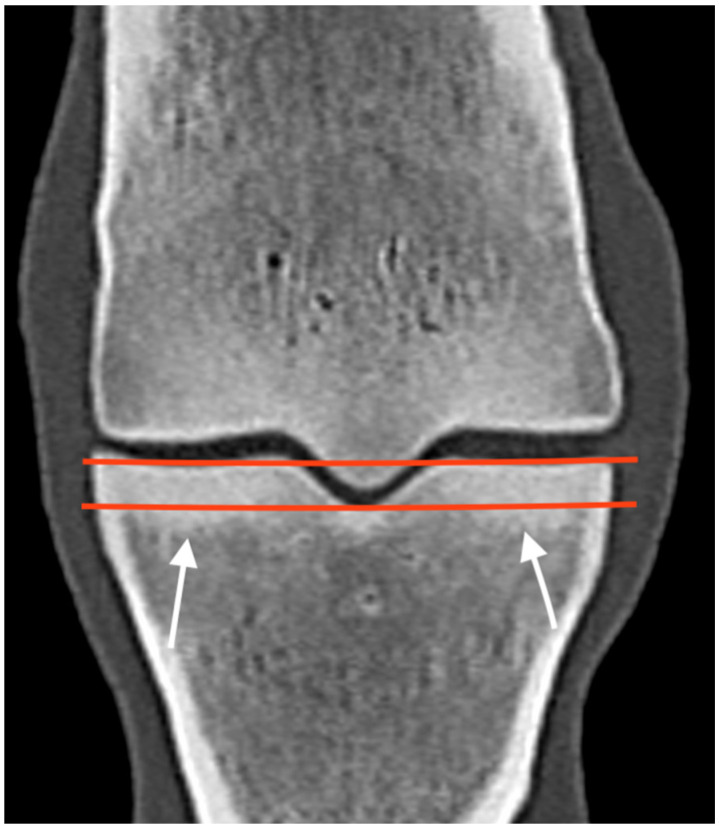
Frontal computed tomographic reconstruction of a metacarpophalangeal joint of a non-lame yearling Thoroughbred. The subchondral bone of the medial and lateral glenoids of the proximal phalanx is thickened (arrows); it extends beyond a line drawn tangential to the distal extent of the articular surface of the sagittal groove, parallel with a line drawn across the articular surface of the medial and lateral glenoids.

**Figure 2 animals-13-03466-f002:**
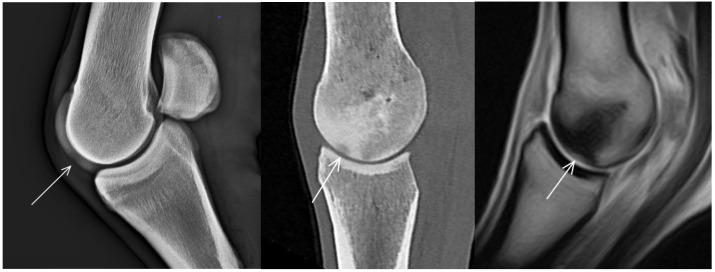
Lateromedial (flexed) radiograph, sagittal computed tomographic (CT) reconstruction and sagittal T1-weighted gradient echo magnetic resonance (MR) image (dorsal is to the left) of a metacarpophalangeal joint of a non-lame Thoroughbred yearling. There is a radiolucent/hypoattenuating/hyperintense lesion in the dorsal subchondral bone of the sagittal ridge of the third metacarpal bone (arrows), suggestive of osteochondrosis. Cone-shaped hyperattenuation/hypointense signal widening towards the lesion is evident in CT and MR images in the dorsal two-thirds of the bone.

**Figure 3 animals-13-03466-f003:**
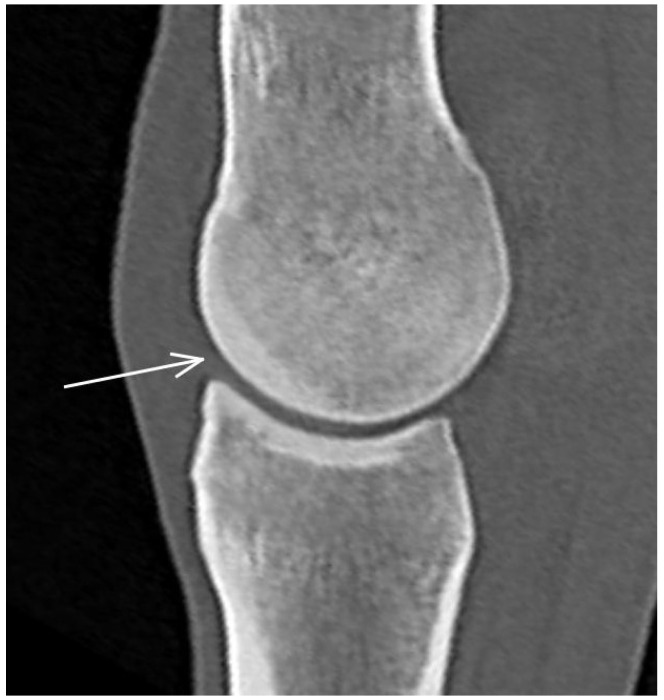
Sagittal computed tomographic reconstruction of a metacarpophalangeal joint of a non-lame Thoroughbred yearling. Dorsal is to the left. There is subchondral bone thickening (arrow) in the dorsal half of the sagittal ridge of the third metacarpal bone.

**Figure 4 animals-13-03466-f004:**
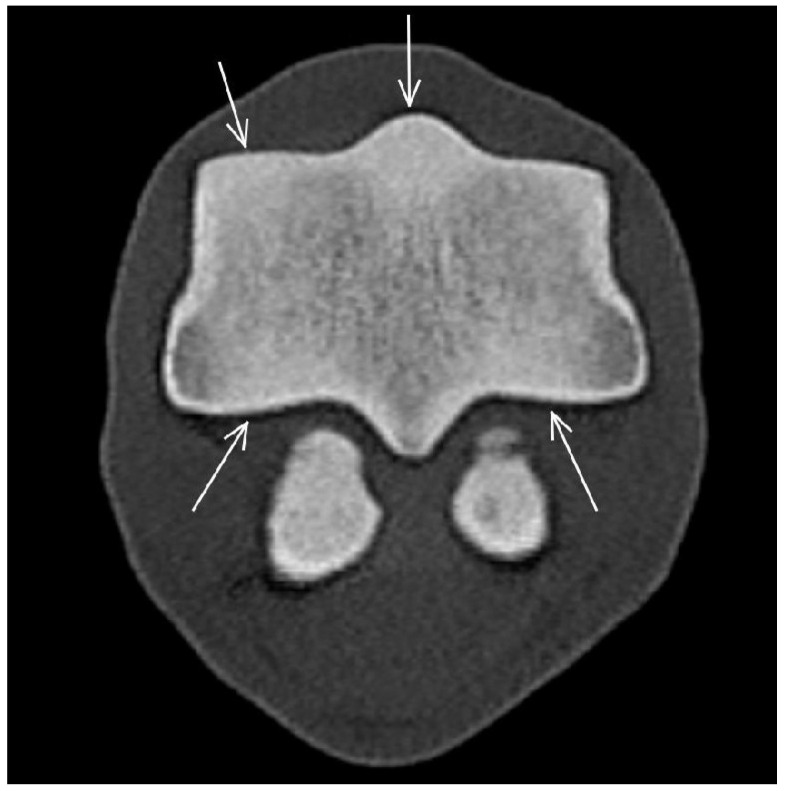
Transverse computed tomographic reconstruction of the distal aspect of the third metacarpal bone of a non-lame Thoroughbred yearling. Medial is to the left. There is mild diffuse hyperattenuation, consistent with increased bone density, in the dorsal aspect of the sagittal ridge and the medial condyle and in the palmar aspect of both condyles (arrows).

**Figure 5 animals-13-03466-f005:**
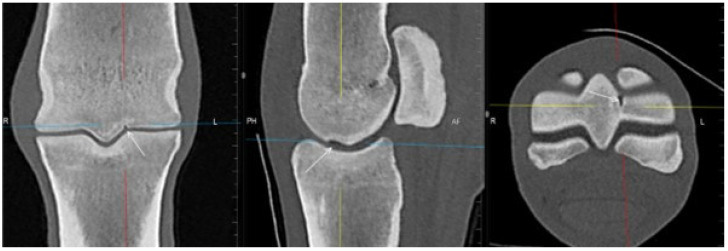
Frontal, sagittal and transverse computed tomographic reconstructions (medial is to the right) of a metacarpophalangeal joint of a non-lame Thoroughbred yearling. There is a smoothly marginated indentation (arrows) in the medial parasagittal groove of the third metacarpal bone. The red, yellow and blue lines indicate the image reconstruction planes.

**Figure 6 animals-13-03466-f006:**
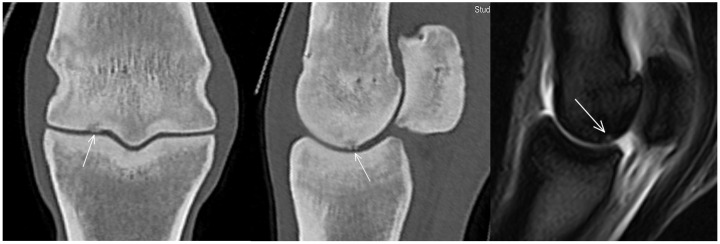
Frontal (medial to the left) and parasagittal computed tomographic (CT) reconstructions and a parasagittal T2*-weighted gradient magnetic resonance image of a metacarpophalangeal joint of a non-lame yearling Thoroughbred. There is a hypoattenuating/hyperintense lesion in the medial condyle of the third metacarpal bone (arrows), surrounded by increased attenuation/decreased signal intensity. In CT images, focal hyperattenuation in the articular cartilage could also be seen. Note the thickened subchondral bone in the proximal phalanx.

**Figure 7 animals-13-03466-f007:**
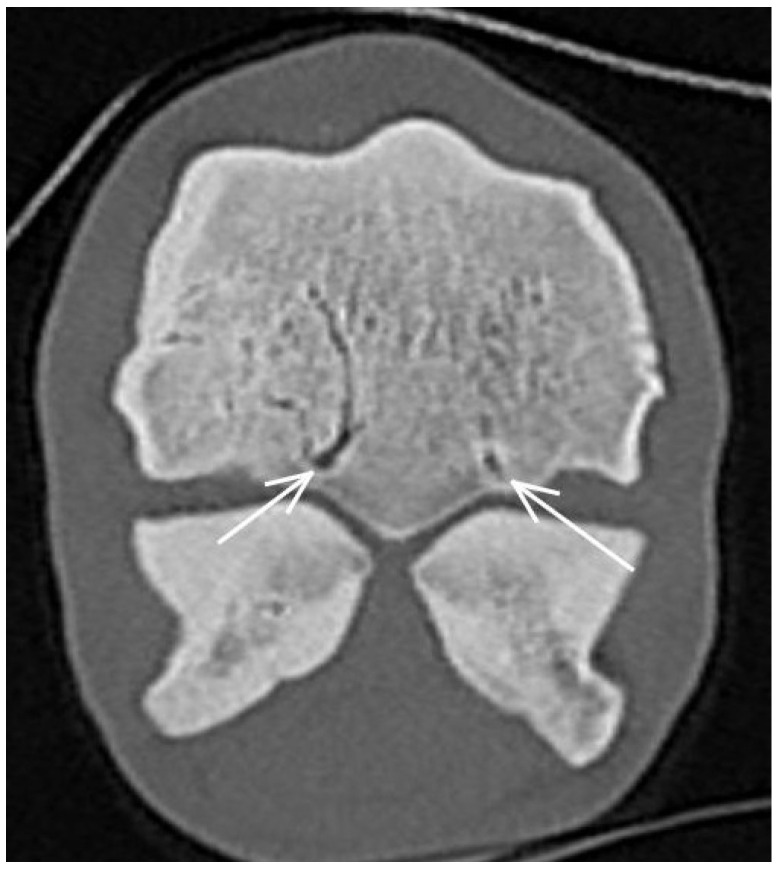
Transverse computed tomographic (CT) reconstruction of a metacarpophalangeal joint of a non-lame Thoroughbred yearling. There are enlarged vascular channels in both metacarpal condyles, originating from the region of the parasagittal grooves (arrows).

**Figure 8 animals-13-03466-f008:**
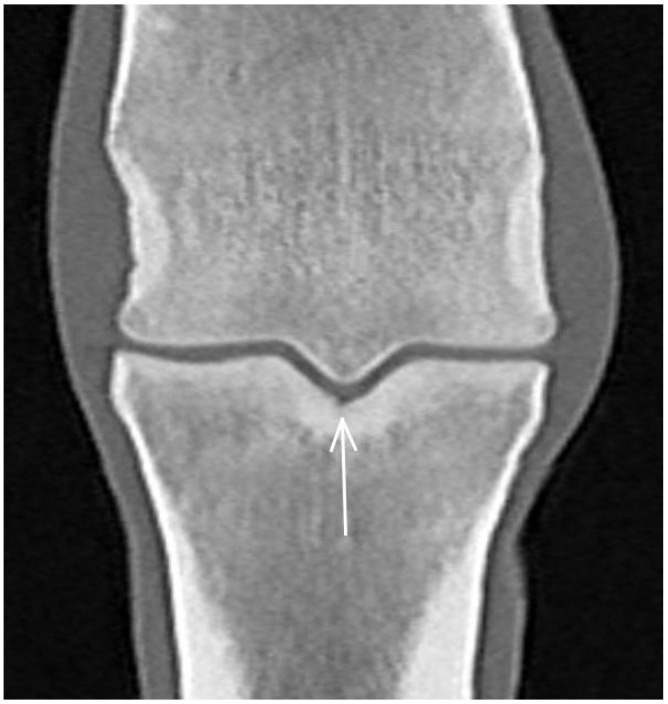
Frontal computed tomographic reconstruction of a metacarpophalangeal joint of a non-lame yearling Thoroughbred. There is a smoothly marginated indentation in the articular surface of the subchondral bone of the sagittal groove of the proximal phalanx (arrow). The subchondral bone in the sagittal groove of the proximal phalanx is thickened.

**Figure 9 animals-13-03466-f009:**
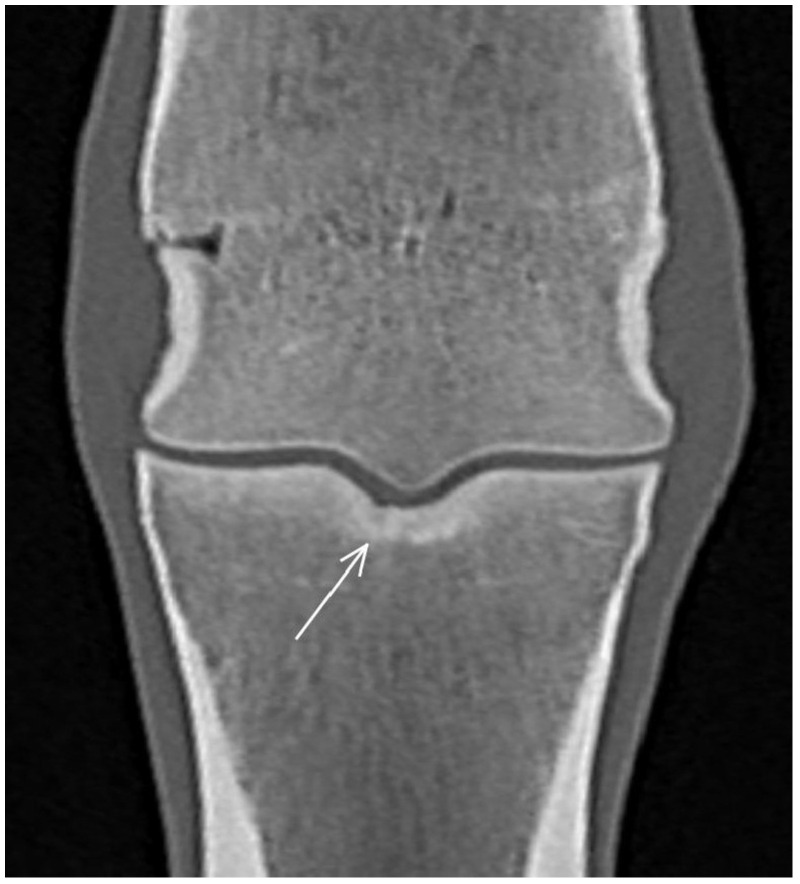
Frontal computed tomographic reconstruction of a metacarpophalangeal joint of a non-lame yearling Thoroughbred. Medial is to the right. There is an irregularly shaped hypoattenuating lesion in the subchondral bone in the lateral aspect of the sagittal groove of the proximal phalanx (arrow). Note the marked difference in thickness of the subchondral bone of the proximal phalanx in the region of the sagittal groove compared with Figure 8.

**Figure 10 animals-13-03466-f010:**
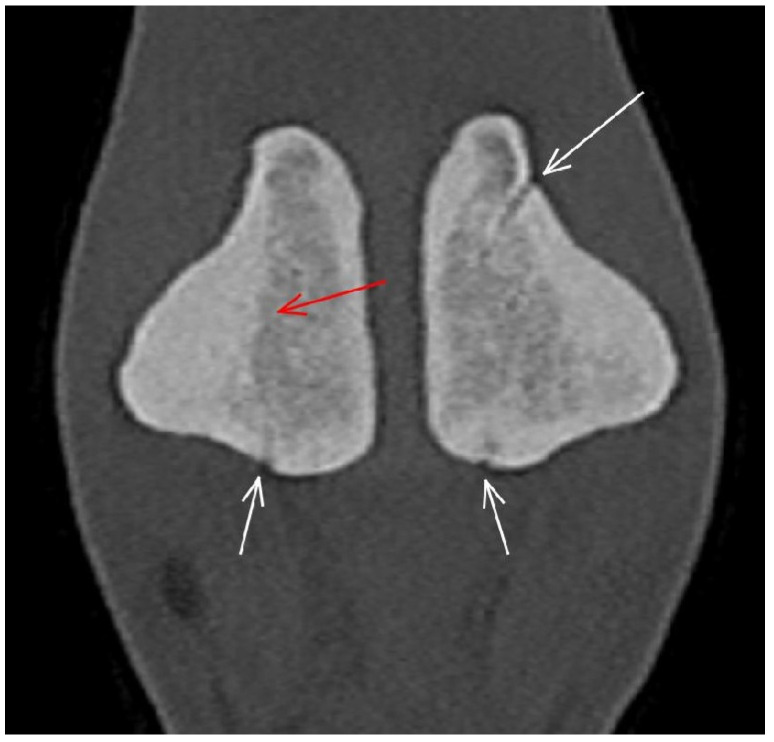
Frontal computed tomographic reconstruction of the proximal sesamoid bones of a forelimb of a non-lame Thoroughbred yearling. Medial is to the left. The abaxial compact bone of the medial proximal sesamoid bone is thickened with hyperattenuation extending into the spongiosa (red arrow). There is a prominent vascular channel in the lateral proximal sesamoid bone (long white arrow), and there are also small vascular channels entering the base of both proximal sesamoid bones (small white arrows).

**Table 1 animals-13-03466-t001:** Pulse sequence parameters used in a 0.27 T magnet to image the metacarpophalangeal joints. The same parameters were used for sagittal, frontal and transverse sequences. T1W GRE—T1-weighted gradient echo, T2*W GRE—T2*-weighted gradient echo, T2W FSE—T2-weighted fast spin echo, STIR FSE—short-tau inversion recovery fast spin echo, MI—motion insensitive, TE—echo time, ms—milliseconds, TR—repetition time, FE—frequency encoding, PE—phase encoding, FOV—field of view.

Pulse Sequence	TE (ms)	TR (ms)	Flip Angle (°)	Slice Thickness (mm)	Slice Gap (mm)	Matrix Size (FE × PE)	FOV (mm)
Pilot	7	66	45	7	0	150 × 120	220
Pilot of a pilot	7	66	45	7	0	150 × 120	220
STIR TEST	22	2100	50/60/85/110	4	0.8	256 × 144	200
T1W GRE MI	8	50	55	5	1	170 × 130	170
T2*W GRE MI	13	68	25	5	1	340 × 160	170
T2W FSE FAST	88	1544	90	5	1	168 × 168	170
STIR FSE FAST	22	2336	95	5	1	168 × 168	170

**Table 2 animals-13-03466-t002:** Description of imaging observations of the metacarpophalangeal joint regions for computed tomography (CT), magnetic resonance imaging (MRI) and radiography in Thoroughbred yearlings. McIII = third metacarpal bone; N/A = not applicable.

Region	CT	MRI	Radiography
McIII Sagittal ridge			
	Hypoattenuating lesion in the dorsal subchondral bone Subchondral bone thickening Dorsal/palmar half Increased attenuation in the trabecular bone Dorsal/palmar half Cone shaped/patchy Focal separated hyperattenuation	Hyperintense signal in the dorsal subchondral bone Subchondral bone thickening Dorsal/palmar half Decreased signal intensity in the trabecular bone Dorsal/palmar half Cone shaped/patchy Focal separated intermediate/low signal intensity	Radiolucent lesion Subchondral bone thickening Dorsal Increased trabecular bone opacity
Medial/lateral condyle			
	Subchondral bone thickening Dorsal/palmar half Increased attenuation in the trabecular bone Dorsal/palmar half Hypoattenuating lesion in the subchondral bone Location	Subchondral bone thickening Dorsal/palmar half Decreased signal intensity in the trabecular bone Dorsal/palmar half Increased signal intensity in the subchondral bone Location	Increased opacity in the trabecular bone Lucent lesion in the subchondral bone Location
Proximal phalanx Sagittal groove			
	Subchondral bone thickening Dorsal/middle/palmar third Increased attenuation in the trabecular bone Hypoattenuating lesion in the subchondral bone Location	Subchondral bone thickening Dorsal/middle/palmar third Decreased signal intensity in the trabecular bone Increased signal intensity in the subchondral bone Location	Subchondral bone thickening Increased opacity in the trabecular bone Lucent lesion in the subchondral bone Location
Medial/lateral glenoid			
	Subchondral bone thickening Dorsal/middle/palmar third Increased attenuation in the trabecular bone Hypoattenuating lesion in the subchondral bone Location Periarticular modelling	Subchondral bone thickening Dorsal/middle/palmar third Decreased signal intensity in the trabecular bone Increased signal intensity in the subchondral bone Location Periarticular modelling	Subchondral bone thickening Increased opacity in the trabecular bone Lucent lesion in the subchondral bone Location Periarticular modelling
Proximal sesamoid bone Medial/Lateral			
	Prominent vascular channels Thickening of axial/abaxial compact bone Increased attenuation in trabecular bone Modelling	Thickening of axial/abaxial compact bone Decreased signal intensity in trabecular bone Modelling	Prominent vascular channels Modelling
Suspensory ligament branches	Asymmetry in size between medial and lateral branches Increased attenuation Enlargement of cross-sectional area	N/A	N/A
Oblique sesamoidean ligaments	Asymmetry in size between medial and lateral ligaments Increased attenuation Enlargement of cross-sectional area	Asymmetry in size between medial and lateral ligaments Increased signal intensity Enlargement of cross-sectional area	N/A
Other soft tissues abnormality	Increased attenuation Enlargement of cross-sectional area Abnormal shape	Increased signal intensity Enlargement of cross-sectional area Abnormal shape	N/A

## Data Availability

Anonymised raw data are available upon reasonable request.

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
