# Peer review of "Magnetic Resonance Imaging, Computed Tomographic and Radiographic Findings in the Metacarpophalangeal Joints of 40 Non-Lame Thoroughbred Yearlings"

_animals, 2023, doi:10.3390/ani13223466_

Round 1
Reviewer 1 Report
Comments and Suggestions for Authors
General comments
The aim of this study is stated to “provide reference data on abnormalities already present in young Thoroughbreds with no prior intensive training”. The comparison of CT, MRI and radiographic findings is important and should contribute to our understanding of bone adaptation to exercise in young TBs. The changes that occur in the subchondral bone of the racehorse fetlock during training are extremely relevant to understanding subchondral bone disease that leads to catastrophic fracture and osteoarthritis, and therefor the data collected in this study is important. The imaging methodology is well explained. The study does provide interesting information that would be useful but the usefulness of the information is limited by a basic flaw in the study design regarding the documentation of the exercise histories of the included horses. This is important for this particular study but also for any subsequent information that will be analysed from this population of horses. It is clear from this manuscript that further publications are planned from the larger longitudinal study being conducted. This longitudinal study will be unique and useful in helping us to understand changes in TB bone with training.
The introduction would benefit from an indication of which regions of the distal MC3 or proximal P1 are of most interest and why. There is literature related to the common finding on radiographs, CT and MRI in horses that have non-adaptive stress related injuries and the reader would benefit from this information in brief. It would put the descriptive results into context.
The materials and methods indicate the selection process for the young TBs that were used in the study. There is a statement that subjective and objective lameness assessment was used, but in the results, the objective lameness assessment is discarded as unreliable since the horses were unable to trot well enough for data to be collected so the objective measurement is not included. This could call into question whether the subjective assessment was also consistent and reliable to detect lameness. In addition, the exercise history is included only as vague detail – saying that the horses had been engaged in “some form of training”. The exercise history is important for evaluation of the findings on the diagnostic imaging since the alterations noted in the CT and MRI as regards areas of increased density on CT and densification on MRI may be directly related to exercise intensity. There seems to be the potential to separate the horses identified as “in some form of training” and for how long, from those that has not undergone training, to compare the findings on CT, MRI and radiographs. This baseline information regarding the study population is not only important for this initial descriptive study but also going forward, to indicate whether the starting population in the longitudinal study was indeed uniform and not different. A lack of evidence that the horses used in the study were similar at the starting point could affect the subsequent interpretation of the results.
At first reading, the results clearly describe the imaging findings in the three modalities that were used. The images presented are clear with succinct descriptions of the findings noted on the sagittal ridge, metacarpal condyles, proximal first phalanx and proximal sesamoid bones; however, the findings are observational. The supplementary material 1 provides descriptions of the criteria used but the table could be simplified (see comments below) for easier understanding of the system used. The advantage of the advanced imaging in CT and MR is the ability to look at specific slices and to better define specific anatomic locations for review and analysis. The authors use the ability to looks at specic areas clearly in the manuscript for looking at the sagittal ridge but is under-utilized for the analysis of metacarpal condyles and proximal P1 and the subchondral bone. The thickness of the SCB could actually be measured on CT and MRI which would provide objective information regarding SCB thickening and the location of this abnormality. In addition, CT could be used to measure bone density. There would be additional work involved in these extra measurements; however, this would also provide more objective information. This reviewer can assume that these studies are in progress but it would seem to make some sense to wait to publish the material until the full analysis has been completed.
As for the comment regarding subchondral bone thickness, are the vascular channels really enlarged? Is there a standard size of vascular channels known to be visible in CT or MRI, and if so, then that should be stated as the standard against a description of enlarged was made.
Discussion and conclusions are vague in many places and although the descriptions are clear the paper does not provide the reader with any clear linking of the conclusions to the exercise histories which is the potential huge benefit of this study set. There are specific comments below regarding the discussion.
The papers from Schaffer and Creswell should be in the references as regards the findings on the proximal sesamoid bones since both these authors have evaluated PSBs with advanced imaging modalities and histology.
Specific comments:
Line |
Simple summary |
15 |
“on” - change to “in”, “on how” – suggest – regarding how |
|
This section may need changing after revision |
|
|
|
Abstract |
34 218 |
Mild trabecular mineralisation – trabecular are mineralised by definition so the meaning of this finding is not clear. Did you mean increased thickness or number of trabeculae. |
38 |
The fact that signs of bone modelling are seen in the images is true but it would be good to link these findings to a clinical relevance. |
|
|
|
Materials and Methods |
69-72 |
The description of the lameness evaluation could be expanded. Which objective lameness tool was used. This is relevant for later when the objective measurement is discarded. |
|
Supplementary material 1 – This chart is somewhat confusing since the formatting does not easily allow understanding of the features evaluated on each modality. Perhaps this is a function of the formatting/something that altered in the transfer of documents to the journal’s site. This table could be simplified for easier understanding since the subjective findings evaluated at the medial and lateral condyles, medial and lateral glenoid and medial and lateral PSBs are similar. A diagram of the regions defined as dorsal/palmar, or dorsal/mid/ and palmar on the images of MC3 and P1 would be helpful. How was subchondral bone thickening assessed – i.e. what was the baseline for SCB that was not thickened. |
|
|
|
Results |
|
No specific comments are made in this section, since with revision the descriptions and the percentages of images/bones with alterations can be described in relation to the different exercise groups. Then the of the relevance of the imaging findings for interpretation will be clearer. |
|
|
|
Discussion |
281-284 |
This paragraph indicates that this work presented is part of a much larger project that followed these horses over time. A great resource, but this should be included at the start in the M&M section to put the work into context. |
285- 293 |
Please specify in this section if the OCD lesions noted in the previous studies were of the mid sagittal ridge, as noted in the images of the current study or if those were at the proximal extend of the sagittal ridge. The lesions in the different locations likely had a different prevalence and also a much different clinical relevance. |
291 303 |
Different – when using this term here and elsewhere in the discussion, please specify “different from what?” |
299-301 |
I am uncertain why the comparison of MR and CT is beyond the scope of this study. The images from both modalities are available and presented in the results? |
311-312 |
This statement indicates that the exercise histories of the included horses were known, so separation of the study population into different groups as regards exercise levels should be possible. |
315 |
What were the “largest differences”. Does this mean that the largest change over time occurred in this age group as regards the micromorphology of the SCB? Please clarify/expand. |
325 |
Refer to previous comment regarding definition of enlarged vascular channel. |
327 |
Endosteal usually refers to the inner surface of cortical bone so the term is a little confusing here. Could this word be deleted and retain the meaning you intend?
|
327-328 |
Could you add why you think the smooth nature of the defect indicates a developmental abnormality? Many OCD lesions have irregular subchondral bone margins. |
331-333 345-347 |
Hopefully your further analysis of data in the longitudinal study will help determine the significance. |
348-351 |
This paragraph on PSBs should include reference to the works of Schaffer and Creswell. |
353 |
Suggest that the sections on soft tissue findings be deleted with expansion of the PSB information – so that this manuscript focuses on the bone findings alone . |
|
|
|
|
|
|
Author Response
We thank the reviewer for their constructive criticism and suggestions. We have addressed each comment and made changes in the manuscript accordingly.
- The aim of this study is stated to “provide reference data on abnormalities already present in young Thoroughbreds with no prior intensive training”. The comparison of CT, MRI and radiographic findings is important and should contribute to our understanding of bone adaptation to exercise in young TBs. The changes that occur in the subchondral bone of the racehorse fetlock during training are extremely relevant to understanding subchondral bone disease that leads to catastrophic fracture and osteoarthritis, and therefor the data collected in this study is important. The imaging methodology is well explained. The study does provide interesting information that would be useful but the usefulness of the information is limited by a basic flaw in the study design regarding the documentation of the exercise histories of the included horses. This is important for this particular study but also for any subsequent information that will be analysed from this population of horses. It is clear from this manuscript that further publications are planned from the larger longitudinal study being conducted. This longitudinal study will be unique and useful in helping us to understand changes in TB bone with training.
Thank you for your comments. Exercise history is available for all horses and in addition to that we also collected data on husbandry (e.g., paddock size, hours spent in paddock) and any exercise prior to training; the latter data are available for 34 horses. However, the aim of this study was to provide descriptive data on imaging findings in young Thoroughbreds entering racehorse training and not to investigate potential association between the horses’ history and imaging findings. We very much agree with the reviewer that this point is interesting and highly important, however it will require careful assessment of the data and objective image analysis that will be performed later. We discussed our descriptive subjective data with statisticians, and they felt very strongly that our case numbers was not enough for robust statistical analysis, especially if it was to be broken down into those horses which had not been ridden and those which had been introduced to light ridden exercise. We therefore opted for descriptive analysis and simply documented that these imaging findings can be present in yearling Thoroughbreds that have not done any fast work. We have added more information on exercise history (lines 133-134), however the main message remains that no horses had done any fast work under saddle (we cannot account for how they self-exercised in fields) and they were all in the first two months of racehorse training. It has been previously demonstrated (e.g. by Martig et al. 2018) that not just the training history, but the pre-training history also contribute to adaptive osseous changes, and therefore splitting up horses into groups based on training exercise history alone could lead to misleading results. Subjectively we did not detect a pattern when comparing imaging findings in horses in training and not yet in training. We have added comments on this in the Limitations.
- The introduction would benefit from an indication of which regions of the distal MC3 or proximal P1 are of most interest and why. There is literature related to the common finding on radiographs, CT and MRI in horses that have non-adaptive stress related injuries and the reader would benefit from this information in brief. It would put the descriptive results into context.
Thank you for the suggestion; commonly affected regions have been added.
- The materials and methods indicate the selection process for the young TBs that were used in the study. There is a statement that subjective and objective lameness assessment was used, but in the results, the objective lameness assessment is discarded as unreliable since the horses were unable to trot well enough for data to be collected so the objective measurement is not included. This could call into question whether the subjective assessment was also consistent and reliable to detect lameness. In addition, the exercise history is included only as vague detail – saying that the horses had been engaged in “some form of training”. The exercise history is important for evaluation of the findings on the diagnostic imaging since the alterations noted in the CT and MRI as regards areas of increased density on CT and densification on MRI may be directly related to exercise intensity. There seems to be the potential to separate the horses identified as “in some form of training” and for how long, from those that has not undergone training, to compare the findings on CT, MRI and radiographs. This baseline information regarding the study population is not only important for this initial descriptive study but also going forward, to indicate whether the starting population in the longitudinal study was indeed uniform and not different. A lack of evidence that the horses used in the study were similar at the starting point could affect the subsequent interpretation of the results.
We fully admit that lameness examination in unhandled yearlings is limited. However, we could confidently say that the included horses did not show overt lameness in a straight line. During subjective lameness assessment we can make a judgement on strides when horses trot in a straight line. The objective lameness assessment tool used in this study requires a minimum of 25 strides in a straight line at a consistent speed, with the head in alignment with the trunk, to provide reliable results. This was simply impossible in the vast majority of horses, as many of them had never been trotted up before. When needed, horses were assessed over the course of several trot ups to conclude the subjective assessment. One horse presented to us was not included in the study due to mild but consistent hindlimb lameness.
With regards to exercise history, please see our comments above.
- At first reading, the results clearly describe the imaging findings in the three modalities that were used. The images presented are clear with succinct descriptions of the findings noted on the sagittal ridge, metacarpal condyles, proximal first phalanx and proximal sesamoid bones; however, the findings are observational. The supplementary material 1 provides descriptions of the criteria used but the table could be simplified (see comments below) for easier understanding of the system used. The advantage of the advanced imaging in CT and MR is the ability to look at specific slices and to better define specific anatomic locations for review and analysis. The authors use the ability to looks at specic areas clearly in the manuscript for looking at the sagittal ridge but is under-utilized for the analysis of metacarpal condyles and proximal P1 and the subchondral bone. The thickness of the SCB could actually be measured on CT and MRI which would provide objective information regarding SCB thickening and the location of this abnormality. In addition, CT could be used to measure bone density. There would be additional work involved in these extra measurements; however, this would also provide more objective information. This reviewer can assume that these studies are in progress but it would seem to make some sense to wait to publish the material until the full analysis has been completed.
Thank you for this comment. The aim of the current paper was to provide descriptive data on findings that clinicians may encounter when reading images of young racehorses. Objective analysis of several regions of interest in each bone is being carried out and results will be published separately because each include a huge amount of data. It should also be noted that measurements in CT and MR images will not be directly comparable. Low-field MRI produces slices of 3-5 mm thickness, acquired in pre-set planes, which cannot be ideally positioned for each measurement in a clinical setting in live horses. The CT used in the current study operates with 0.5 mm slice thickness in the distal limb and images can be reconstructed in any plane, which allows much more accurate measurement of e.g., subchondral bone thickness.
- As for the comment regarding subchondral bone thickness, are the vascular channels really enlarged? Is there a standard size of vascular channels known to be visible in CT or MRI, and if so, then that should be stated as the standard against a description of enlarged was made.
To our knowledge, definition has only been provided for radiographic interpretation, and that has varied somewhat among published studies. We considered vascular channels to be enlarged in CT images when they were more prominent than just being visible. Objective grading system and measurements on both vascular channel size and bone density will be performed in a separate study focussing on the proximal sesamoid bones.
- Discussion and conclusions are vague in many places and although the descriptions are clear the paper does not provide the reader with any clear linking of the conclusions to the exercise histories which is the potential huge benefit of this study set. There are specific comments below regarding the discussion.
Please see our comments above. It was not the aim of the paper to relate imaging findings to exercise history, but to document imaging findings that can be present in yearling Thoroughbreds entering racehorse training. The longer term objective of the entire longitudinal study will relate cumulative work with imaging findings.
- The papers from Schaffer and Creswell should be in the references as regards the findings on the proximal sesamoid bones since both these authors have evaluated PSBs with advanced imaging modalities and histology.
Thank you; references have been added.
Specific comments:
Line |
Simple summary |
15 |
“on” - change to “in”, “on how” – suggest – regarding how |
|
This section may need changing after revision |
|
Changed, thank you. |
|
Abstract |
34 218 |
Mild trabecular mineralisation – trabecular are mineralised by definition so the meaning of this finding is not clear. Did you mean increased thickness or number of trabeculae. Changed to ‘hyperattenuation’. |
38 |
The fact that signs of bone modelling are seen in the images is true but it would be good to link these findings to a clinical relevance. In the Abstract we are limited by the word count allowance; the clinical relevance is discussed in the Discussion. |
|
|
|
Materials and Methods |
69-72 |
The description of the lameness evaluation could be expanded. Which objective lameness tool was used. This is relevant for later when the objective measurement is discarded. Information has been added. |
|
Supplementary material 1 – This chart is somewhat confusing since the formatting does not easily allow understanding of the features evaluated on each modality. Perhaps this is a function of the formatting/something that altered in the transfer of documents to the journal’s site. This table could be simplified for easier understanding since the subjective findings evaluated at the medial and lateral condyles, medial and lateral glenoid and medial and lateral PSBs are similar. A diagram of the regions defined as dorsal/palmar, or dorsal/mid/ and palmar on the images of MC3 and P1 would be helpful. How was subchondral bone thickening assessed – i.e. what was the baseline for SCB that was not thickened. Thank you for the comments. When we download the supplementary material, we get the original format, so we cannot comment on why it looks confusing. Except for the radiographs, assessment was not done on a single image but on multiple images / reconstructions, we therefore feel that adding additional images to the Supplementary material would not provide any additional benefits over the figures in the main text that illustrate most findings. More information on subchondral bone thickness assessment has been added to the Materials and Methods (lines 130-132). |
|
|
|
Results |
|
No specific comments are made in this section, since with revision the descriptions and the percentages of images/bones with alterations can be described in relation to the different exercise groups. Then the of the relevance of the imaging findings for interpretation will be clearer. |
|
As explained above, we could not perform such an analysis in a statistically robust way. Any such analysis would be underpowered and the suggested breakdown of the data could lead to misleading interpretation of our results. Moreover it is impossible to quantify the self-regulated exercise that horses may have performed during turn out on a variety of terrains and surfaces. |
|
Discussion |
281-284 |
This paragraph indicates that this work presented is part of a much larger project that followed these horses over time. A great resource, but this should be included at the start in the M&M section to put the work into context. Thank you for the comment; this paragraph has been moved to the beginning of Materials and Methods. |
285- 293 |
Please specify in this section if the OCD lesions noted in the previous studies were of the mid sagittal ridge, as noted in the images of the current study or if those were at the proximal extend of the sagittal ridge. The lesions in the different locations likely had a different prevalence and also a much different clinical relevance. The requested information has been added. |
291 303 |
Different – when using this term here and elsewhere in the discussion, please specify “different from what?” Corrected. |
299-301 |
I am uncertain why the comparison of MR and CT is beyond the scope of this study. The images from both modalities are available and presented in the results? We believe that for adequate comparison of CT and MRI findings, detailed objective analysis is required, which will be performed for each region of interest (bone) separately. |
311-312 |
This statement indicates that the exercise histories of the included horses were known, so separation of the study population into different groups as regards exercise levels should be possible. Please see our comments above. |
315 |
What were the “largest differences”. Does this mean that the largest change over time occurred in this age group as regards the micromorphology of the SCB? Please clarify/expand. Additional information has been added (line 330). |
325 |
Refer to previous comment regarding definition of enlarged vascular channel. Please see our previous response. |
327 |
Endosteal usually refers to the inner surface of cortical bone so the term is a little confusing here. Could this word be deleted and retain the meaning you intend? Endosteal has been deleted. |
327-328 |
Could you add why you think the smooth nature of the defect indicates a developmental abnormality? Many OCD lesions have irregular subchondral bone margins. ‘Developmental’ has been changed to ‘congenital’. |
331-333 345-347 |
Hopefully your further analysis of data in the longitudinal study will help determine the significance. |
348-351 |
This paragraph on PSBs should include reference to the works of Schaffer and Creswell. Added. |
353 |
Suggest that the sections on soft tissue findings be deleted with expansion of the PSB information – so that this manuscript focuses on the bone findings alone . We would prefer to keep soft tissue findings. Detailed information of PSB’s, including follow-up examinations and detailed objective measurements will be published separately. |
Reviewer 2 Report
Comments and Suggestions for Authors
Dear authors,
It is a very interesting study and the manuscript is very well presented. I have no specific suggestions or comments. I only wonder that it would be nice to follow up on those horses that presented changed not only on the bone but also on the suspensory ligament branches and oblique sesamoidean ligaments, and see if in the future they develop or not related injuries.
Kind regards
Author Response
Thank you for the comments. Of course, follow-up assessments will include the soft tissues as well.
Reviewer 3 Report
Comments and Suggestions for Authors
Manuscript: animals- 2652628
line 71-72: Could you please add details about subjective and objective lameness assessment which has been used in your study? Have lameness location devices been used? Has AAEP scale been applied? Clarify and add appropriate references.
Discussion: Recent studies support the use of computed tomographic arthrography (CTA) in order to detect articular cartilage defects, which has displayed high sensitivity compared to magnetic resonance arthrography, MRI, and CT (Suarez Sanchez-Andrade et al. 2017; Hontoir et al. 2014). Do you consider that CTA might have improved your findings? Might it be considered a limit of the present study? Please, consider enforcing your discussion by adding observations on this topics
Author Response
- line 71-72:Could you please add details about subjective and objective lameness assessment which has been used in your study? Have lameness location devices been used? Has AAEP scale been applied? Clarify and add appropriate references.
No grading was used, because only non-lame horses were included in the study. Information on the objective system for lameness assessment has been added and the sentence describing inclusion criteria has been expanded. The first 40 horses that did not show overt lameness in walk or trot were selected.
- Discussion:Recent studies support the use of computed tomographic arthrography (CTA) in order to detect articular cartilage defects, which has displayed high sensitivity compared to magnetic resonance arthrography, MRI, and CT (Suarez Sanchez-Andrade et al. 2017; Hontoir et al. 2014). Do you consider that CTA might have improved your findings? Might it be considered a limit of the present study? Please, consider enforcing your discussion by adding observations on this topics
Thank you for the comment. Although it would have been interesting to confirm the presence of cartilage lesions, it is highly unlikely that cartilage lesions in non-lame yearlings without any clinical sign of metacarpophalangeal joint disease would have been detected. Moreover, performing arthrocentesis in clinically normal horses would be of ethical concern, especially in some relatively unhandled young horses. We are already over the suggested word count and therefore would prefer to discuss these imaging techniques in our subsequent papers where abnormalities potentially associated with cartilage damage are performed.
Reviewer 4 Report
Comments and Suggestions for Authors
Dear authors,
I have read with much interest your paper and must compliment you on the excellent iconography you have produced and with which you describe very well the CT/RM/RX findings of subdle fetlock lesions of young track horses not yet in training. Below you will find my comments line by line, first however, I would like to elaborate more on your motivations for doing this study. Considering that we are dealing with young horses that, although they do not show clinical alterations, still have potentially relevant bone lesions. Would you recommend such investigations as screening before starting actual training? What influence might the discovery of such alterations have on the sports management of these horses? Would you have them discarded? Would you suggest waiting before mattering them in training? In general, what kind of suggestions might this article make to veterinarians who treat racehorses?
I also believe, as explained in detail below, that there is confusion in the use of certain terms. Using synonyms or different terms to describe the same alteration leads to confusion in the reader, especially when even in the scientific literature there is no consesus regarding the nomenclature of specific pathologies.
Line 33: In the paper, the authors often refer to “trabecular mineralization” or “hyperattenuation in trabecular bone”. In literature, however, other authors prefer to use terms as “bone sclerosis” or “increased trabecular bone density” or “increased bone density”. As you know, for years, terminologies such as bone bruise, bone oedema or bone contusion were used interchangeable, generating confusion in the reader until the generic term bone marrow lesion or BML was agreed upon. Therefore, I believe that even for the MRI finding described by the authors and referred to as "hyperattenuation in trabecular bone," it is more appropriate to use a commonly used and more specific term.
I will preface this by saying that I do not have a solution but only suggestions. I think the term "bone sclerosis" may be appropriate, as it defines the lesion as an "abnormal increase in density and hardening of bone," although in human medicine it tends to be used for bone diseases such as Paget's disease, hyperparathyroidism, ostepopetrosis, etc. On the other hand, this term could be confusing to the reader and I think it should be used more properly for histopathological description. I strongly disagree with the use of the term "mineralization" when referring to bone trabeculae, since these are physiologically mineralized structures. Instead, I agree with the authors when they describe this alteration as "hyperattenuation in trabecular bone" or, wanting to simplify things, it would be desirable to speak of "bone densification."
I obviously leave the authors free to use the terminology they deem most appropriate with the only recommendation being to be consistent in its use throughout the paper to avoid confusing the reader.
Line 19: Not clear. In this sentence reference is made only to the hypoattenuating lesions of the dorsal aspect of the sagittal ridge. In general, the only changes observed even on radiographic examination concern this particular alteration and no others. Would it perhaps be worth discussing this aspect/limitation of the radiographic examination in the discussions?
Line 33: How do you explain that you can see the cone shape mineralisation of the RSM better in MRI than in CT? Line 154 also states that CT was able to diagnose mild subchondral bone hypsis in 74 out of 80 fetuses
Line 77-89: you refers to low field MRI but I think it is mandatory to specify that examinations have been performed with a standing system. It is very important especially when you define "motion artifacts" as a major limitation of the low filed system. CT have been performed with horses under general anesthesia?
Line 80: I would suggest the term “dorsal” instead “frontal” when referring to the “coronal plane”. But, one time more, is up to you.
Line 124: Did you evaluate a possible a relationship between the greater degree of bone densification and initiation of training? Was the degree of bone densification greater in the population of subjects (27) who had already started training?
Line 178: Talking about hypoattenuating lesion in this case might cause confusion. I think it would be better to use the same terminology as in the caption: “smoothly marginated indentation”
Line 181-185: The alteration described in Fig. 5 could be a finding compatible with Central Osteophytes (COs) as described by Olive et al. (Imaging and histological features of central subchondral osteophytes in racehorses with metacarpophalangeal joint osteoarthritis. Equine Vet J. 2009 Dec;41(9):859-64) and by McIlwraith CW, Frisbie DD, Kawcak CE(The horse as a model of naturally occurring osteoarthritis. Bone Joint Res. 2012 Nov 1;1(11):297-309). Please consider this suggestion and eventually describe it accordingly adding these references
Line 231: Subchondral thickening is mentioned in the text in reference to fig 8. Comparing fig 7 and fig 8 the most thickened subchondral plate appears to be that of the sagittal groove in fig 7 not in the fig 8 as stated
Line 232: It would have been interesting to have data on the conformation of the subjects as well, or do you think this has no bearing on the different distribution of bone densification patterns and/or the presence of asymmetries at ligamentous structures that you found? (e.g., suspensory branches and OSL ). Could this be a point of discussion?
Line 287-293. The paper you refer to with respect to the higher prevalence of radiolucent lesions observed at sagittal ridge by Kane et al. is 20 years old, and the published radiographs are undoubtedly of much lower quality than those used in your study. This in my opinion may have influenced the detection of lesions that are less obvious and/or not associated with changes in bone profile.
Author Response
We thank the Reviewer for their comments and constructive criticism. We have amended the manuscript accordingly.
- I have read with much interest your paper and must compliment you on the excellent iconography you have produced and with which you describe very well the CT/RM/RX findings of subdle fetlock lesions of young track horses not yet in training. Below you will find my comments line by line, first however, I would like to elaborate more on your motivations for doing this study. Considering that we are dealing with young horses that, although they do not show clinical alterations, still have potentially relevant bone lesions.
Thank you for your comments.
- Would you recommend such investigations as screening before starting actual training? What influence might the discovery of such alterations have on the sports management of these horses? Would you have them discarded? Would you suggest waiting before mattering them in training? In general, what kind of suggestions might this article make to veterinarians who treat racehorses?
We do not believe that we have sufficient information to make such recommendations. For now, our results can help clinicians who evaluate diagnostic imaging results of young Thoroughbreds and hopefully help avoid overinterpretation of results. For risk assessment and recommendations concerning future training follow-up data will be essential.
- I also believe, as explained in detail below, that there is confusion in the use of certain terms. Using synonyms or different terms to describe the same alteration leads to confusion in the reader, especially when even in the scientific literature there is no consesus regarding the nomenclature of specific pathologies.
- Line 33: In the paper, the authors often refer to “trabecular mineralization” or “hyperattenuation in trabecular bone”. In literature, however, other authors prefer to use terms as “bone sclerosis” or “increased trabecular bone density” or “increased bone density”. As you know, for years, terminologies such as bone bruise, bone oedema or bone contusion were used interchangeable, generating confusion in the reader until the generic term bone marrow lesion or BML was agreed upon. Therefore, I believe that even for the MRI finding described by the authors and referred to as "hyperattenuation in trabecular bone," it is more appropriate to use a commonly used and more specific term.
We agree that consensus is ideal.
I will preface this by saying that I do not have a solution but only suggestions. I think the term "bone sclerosis" may be appropriate, as it defines the lesion as an "abnormal increase in density and hardening of bone," although in human medicine it tends to be used for bone diseases such as Paget's disease, hyperparathyroidism, ostepopetrosis, etc. On the other hand, this term could be confusing to the reader and I think it should be used more properly for histopathological description. I strongly disagree with the use of the term "mineralization" when referring to bone trabeculae, since these are physiologically mineralized structures. Instead, I agree with the authors when they describe this alteration as "hyperattenuation in trabecular bone" or, wanting to simplify things, it would be desirable to speak of "bone densification."
I obviously leave the authors free to use the terminology they deem most appropriate with the only recommendation being to be consistent in its use throughout the paper to avoid confusing the reader.
Thank you for your thoughts and recommendation. We agree that a consensus should be reached to aid discussion of findings. For this manuscript, we are happy to refer to increased opacity / decreased signal intensity / hyperattenuation that reflects increased bone density.
- Line 19: Not clear. In this sentence reference is made only to the hypoattenuating lesions of the dorsal aspect of the sagittal ridge. In general, the only changes observed even on radiographic examination concern this particular alteration and no others. Would it perhaps be worth discussing this aspect/limitation of the radiographic examination in the discussions?
Please note that this is the simple summary that should avoid going into too much technical details. We are not clear what the reviewer suggests we should discuss in the Discussion.
- Line 33: How do you explain that you can see the cone shape mineralisation of the RSM better in MRI than in CT? Line 154 also states that CT was able to diagnose mild subchondral bone hypsis in 74 out of 80 fetuses
The difference might be related to slice thickness and pixel/voxel size. The pattern might be accentuated in MR images where signals are averaged within the 5 mm slice thickness. The CT images were acquired with 0.5 mm slice thickness and reconstructed with 0.3 mm voxel size. It is possible that small changes in attenuation/bone density on single thinner slices / reconstructions are more difficult to visually appreciate than on images where signals / attenuation from a greater slice thickness is averaged. With regards to the second sentence, we are not clear what the Reviewer means.
- Line 77-89: you refers to low field MRI but I think it is mandatory to specify that examinations have been performed with a standing system. It is very important especially when you define "motion artifacts" as a major limitation of the low filed system. CT have been performed with horses under general anesthesia?
In these lines we refer to all three imaging modalities, which has been made clearer. All three modalities were performed with the horses in a standing position.
- Line 80: I would suggest the term “dorsal” instead “frontal” when referring to the “coronal plane”. But, one time more, is up to you.
Frontal is the standard terminology used for orientation of the coronal plane in the low-field MRI system used in this study and therefore we would prefer to keep the text as it is.
- Line 124: Did you evaluate a possible a relationship between the greater degree of bone densification and initiation of training? Was the degree of bone densification greater in the population of subjects (27) who had already started training?
This would have been very interesting, we agree. We discussed our descriptive subjective data with statisticians, and they felt very strongly that our case number was not enough for robust statistical analysis separating those horses which had started light ridden work compared with those which had not. Moreover, it is impossible to quantify the self-regulated exercise that horses may have performed during turn out on a variety of terrains and surfaces. We therefore opted for descriptive analysis and simply document that these imaging findings can be present in yearling Thoroughbreds that have not done any fast work. We have added more information on exercise history (lines 133-134), however the main message remains that no horses had done any fast work and they were all in the first two months of racehorse training. It has been previously demonstrated (e.g. by Martig et al. 2018) that not just the training history, but the pre-training history also contribute to adaptive osseous changes, and therefore splitting up horses into groups based on training exercise history alone could lead to misleading results. Subjectively we did not detect a pattern when comparing imaging findings in horses in training and not yet in training. We have added comments on this in the Limitations.
- Line 178: Talking about hypoattenuating lesion in this case might cause confusion. I think it would be better to use the same terminology as in the caption: “smoothly marginated indentation”
Not all were smoothly marginated, but we agree, lesion is probably not the best terminology. We have replaced it with ‘region’.
- Line 181-185: The alteration described in Fig. 5 could be a finding compatible with Central Osteophytes (COs) as described by Olive et al. (Imaging and histological features of central subchondral osteophytes in racehorses with metacarpophalangeal joint osteoarthritis. Equine Vet J. 2009 Dec;41(9):859-64) and by McIlwraith CW, Frisbie DD, Kawcak CE(The horse as a model of naturally occurring osteoarthritis. Bone Joint Res. 2012 Nov 1;1(11):297-309).Please consider this suggestion and eventually describe it accordingly adding these references
Thank you for the suggestion, a paragraph has been added to the Discussion citing Olive et al.. The McIlwraith et al. review paper refers to the Olive study and does not seem to add further information.
- Line 231: Subchondral thickening is mentioned in the text in reference to fig 8. Comparing fig 7 and fig 8 the most thickened subchondral plate appears to be that of the sagittal groove in fig 7 not in the fig 8 as stated
Thank you, this has been corrected.
Round 2
Reviewer 1 Report
Comments and Suggestions for Authors
Author Response
Response to Reviewer 1
We thank for the Reviewer’s further comments. We have addressed each suggestion and query.
1. I would still suggest that in this descriptive study that a quick look for any descriptive differences in the appearance of the images of horses with more or less training would be beneficial (especially on the MRI and CT images). One of the really positive aspects of this study will be to help us understand what alterations in CT and MRI images noted are physiologic (normal response to training) and which are pathological. Only by incorporating training details will this be possible and it should start with this first study- since any differences in the horses at the start will influence the further studies that are underway. For example, do we know that the radiolucent/hypoattenuated areas in the sagittal ridge and sagittal groove.
As discussed following the first review, we do not think that grouping this relatively small population according to subjective imaging findings and training history would provide valid results. We will look into training and husbandry data when we have objective results on bone density (Mean Hounsfield Unit values will be obtained in different parts of McIII).
2. The supplementary material 1 could easily be made clearer with the addition of some bullets to indicate a separation of the different subjective/observational criteria that were used. The medial and lateral measurements appear to be the same for MC3 and Pl and the PSBs so the table could be simplified by combining these and making the header "medial and lateral". lf the table were more succinct then it could be included as a table in the manuscript and the reader would have a clearer idea of what was evaluated.
Thank you for the suggestion, the Table is now shorter and inserted in the main text as Table 2.
3. The challenge of matching CT and MRI slices is appreciated by the reviewers. However, there should be a means in the 3D imaging modalities with different planes of view to make a good effort to closely match slices .. such as at 25 %, 50% etc of the curve of theMC3 condyles or at 25, 50 or 75% of the distance from dorsal to palmar of the Pl glenoids.
Valid direct comparison would only work if isometric MRI sequences were available, which, unfortunately is not a practical option when acquiring MR images of most horses in a standing position in the low-field system. Computed tomographic images can be reconstructed perpendicular to the articular surface, but in clinical settings MR image are not obtained in multiple frontal views to produce images set perpendicular to the articular surface of various parts of McIII and the proximal phalanx. Once we have data available on the follow up examinations, with a larger number of imaging studies we will work on the best possible system to compare CT and MR images.
Specific comments
4. Abstract - suggest removing from line 28 - "how the lesions progress" since that is not studied in this manuscript and it is misleading.
Removed
5. Line 68-70- indicate that these are Thoroughbred racehorses to be followed from start of training to race fitness. Current wording is vague regarding level of fitness.
Clarification has been added.
6. Line 117- 119 - this is still not clear how SCB thickness was determined. Please try to reword for clarity.
We agree that this was not clear. Further information and a figure have been added for clarification.
7. Line 193-195 If the enlarged vascular channels are present in 60-70% of the limbs then one could argue that these are normal, if seen in so many individuals. This relates to my previous comments about determining "enlarged" especially with the use of new technologies in new ways. In the discussion, it would be good to consider this interpretation that the "enlargement" could be normal and also that is could be related to our better ability to appreciate detail in CT.
Prominent vascular channels were detected also radiographically in 92.5% of limbs, based on previously published guidelines for what is considered normal in young Thoroughbreds. While it may be incidental, we do not consider this a normal variation. If we compare CT and radiography results, in this particular finding there was not a big difference (100% vs. 92.5%). Computed tomography helped identify thickening of the compact bone, which was not visible on radiographs.
8. Line 320 - 321 This may not be a logical comparison. The previous study cited is a longitudinal study and your is observational and a snapshot at one time so far. So maybe use this to say conflicting results, but the reason unknown and the longitudinal aspect of the current study may facilitate understanding if alterations are pathological or physiological?
Thank you for the suggestion. We have added a sentence stating that results of the two studies are not directly comparable.
‘Moreover, we do not have information on bone density in the study population at an earlier age, therefore results of the two studies are not directly comparable.’
9. Line 326-329 How are the differences in micromorphology similar to the current study? Not clear.
Thank you for the suggestion, clarification has been added to the text.